# Adult and Elderly Risk Factors of Mortality in 23,614 Emergently Admitted Patients with Rectal or Rectosigmoid Junction Malignancy

**DOI:** 10.3390/ijerph19159203

**Published:** 2022-07-27

**Authors:** Lior Levy, Abbas Smiley, Rifat Latifi

**Affiliations:** 1School of Medicine, New York Medical College, Valhalla, NY 10595, USA; llevy6@student.nymc.edu; 2Westchester Medical Center, New York Medical College, Valhalla, NY 10595, USA; abbaset4@gmail.com; 3Department of Surgery, University of Arizona, Tucson, AZ 85721, USA

**Keywords:** malignant neoplasm of rectum and rectosigmoid junction, in-hospital mortality, hospital length of stay

## Abstract

Background: Colorectal cancer, among which are malignant neoplasms of the rectum and rectosigmoid junction, is the fourth most common cancer cause of death globally. The goal of this study was to evaluate independent predictors of in-hospital mortality in adult and elderly patients undergoing emergency admission for malignant neoplasm of the rectum and rectosigmoid junction. Methods: Demographic and clinical data were obtained from the National Inpatient Sample (NIS), 2005–2014, to evaluate adult (age 18–64 years) and elderly (65+ years) patients with malignant neoplasm of the rectum and rectosigmoid junction who underwent emergency surgery. A multivariable logistic regression model with backward elimination process was used to identify the association of predictors and in-hospital mortality. Results: A total of 10,918 non-elderly adult and 12,696 elderly patients were included in this study. Their mean (standard deviation (SD)) age was 53 (8.5) and 77.5 (8) years, respectively. The odds ratios (95% confidence interval, P-value) of some of the pertinent risk factors for mortality for operated adults were 1.04 for time to operation (95%CI: 1.02–1.07, *p* < 0.001), 2.83 for respiratory diseases (95%CI: 2.02–3.98), and 1.93 for cardiac disease (95%CI: 1.39–2.70), among others. Hospital length of stay was a significant risk factor as well for elderly patients—OR: 1.02 (95%CI: 1.01–1.03, *p* = 0.002). Conclusions: In adult patients who underwent an operation, time to operation, respiratory diseases, and cardiac disease were some of the main risk factors of mortality. In patients who did not undergo a surgical procedure, malignant neoplasm of the rectosigmoid junction, respiratory disease, and fluid and electrolyte disorders were risk factors of mortality. In this patient group, hospital length of stay was only significant for elderly patients.

## 1. Introduction

Colorectal cancers are the third most diagnosed cancer in males and the second in females [1,2], and the third leading cause of cancer death in men and women in the United States [3]. Rectal cancer is one of the frequent human malignant neoplasms and the second most common cancer in the large intestine [3]. Differentiation between rectal and sigmoid carcinomas is a diagnostic challenge with important implications for further treatment [4]. All tumors from a 0 to 15 cm distance from the anal verge are usually defined as rectal carcinomas, and all tumors more than 15 cm as sigmoid carcinomas [5]. Rectal carcinomas with positive lymph nodes and/or threatened resection margins on MRI are treated with preoperative therapy, but sigmoid carcinomas are not. Therefore, a correct diagnosis made during the pre-treatment workup is vital [6]. Environmental and genetic factors can affect the likelihood of colon and rectal cancers [7]. Important risk factors of both rectosigmoid junction and rectal cancers are, for example, age, sex, BMI, diet, genetic predisposition, and physical activity. Mortality rate is 30–40% higher in men than in women, though this difference varies by age. Race and ethnicity can also affect the mortality rate; for instance, recent reports from the United States show death rates in Blacks are more than double those in Asians/Pacific Islanders [3]. Studies have shown that patients with rectosigmoid junction neuroendocrine tumors have a better survival and different risk factors than those with rectal neuroendocrine tumors. The treatment choices for rectal neuroendocrine tumors and rectosigmoid junction neuroendocrine tumors may need to be reconsidered [8]. In a study that investigated the effect of treatment delay on cancer-related outcomes in a large, continuous series of surgically treated colon cancer patients, Amri et al. have shown that the delay of treatment was significantly related to the total length of hospital stay, increased morbidity, and mortality [9]. Prolonged time to operation due to the COVID-19 pandemic has shown to be associated with shorter survival times in colorectal cancer [10]. Risk factors for adverse outcomes following emergency surgery for rectal and rectosigmoid neoplasm complications are still debated. The aim of this study was to evaluate the predictors of in-hospital mortality following emergency surgery for complicated rectal and rectosigmoid neoplasms to help identify ways to improve the field and achieve better patient outcomes.

## 2. Materials and Methods

The National Inpatient Sample (NIS) is a database that is part of the Healthcare Cost and Utilization Project (HCUP), a project sponsored by the Agency for Healthcare Research and Quality (AHRQ). With an annual broad reach of an estimated 7 million patient records, the NIS provides a great degree of power of data analysis across many domains, such as age, sex, clinical characteristics, and geographical location within the United States. The NIS employs the process of weighting when creating the sample of discharges from community hospitals in the US, excluding rehabilitation centers and long-term acute care facilities. This method of stratification allows national estimates of hospitalizations to be made for certain factors. This retrospective cohort study extracted data with the following inclusion criteria: (1) non-elderly adult patients (ages 18–64 years) and elderly patients (65+) (2) with a malignant neoplasm of the rectum or of the rectosigmoid junction (3) who underwent emergency admission from NIS, 2005–2014. The ICD-9 code used to identify patients with a malignant neoplasm of the rectum and rectosigmoid junction was 154. Table 1 contains the ICD-9 codes for surgeries and invasive diagnostic and therapeutic procedure data. The following characteristics of patients and hospitals were collected and analyzed: age, sex, race, income quartile, primary diagnosis, health care insurance (Medicare, Medicaid, private insurance, self-paid, and no charge), hospital location (rural, urban: non-teaching, urban: teaching), neoplasm location (rectosigmoid junction, rectum), invasive diagnostic and/or therapeutic procedure status, surgical procedure status, hospital length of stay, and total charges.

### Statistical Analysis

Descriptive statistics were utilized to express categorical variables as numbers, percentages, and ratios. Continuous variables were presented as means and standard deviation. The normality of data was tested through histograms and the Kolmogorov–Smirnov test to make sure that it followed a normal distribution. Given the very large sample size, small departures from normality did not preclude further statistical analysis. Any data that were not normal were examined for outliers to remove from the distribution. If there were no outliers, the data could also undergo a transformation, such as a log or square root to make it normal. Chi square and Student’s t tests were used to compare categorical and continuous variables, respectively. Independent variables were stratified in three ways: (1) according to sex and either adult or elderly, (2) survived patients vs. deceased ones within each age group, and (3) had an operation or did not for both adults and elderly patients. The dependent variable was mortality. The same stratifications were applied to compare the mortality between men and women, deceased vs. survived patients, and operated vs. not-operated patients. Binary multivariable logistic regression analyses with backward elimination were adjusted for the following characteristics of patients and hospitals: age, sex, race, income quartile, health care insurance, hospital location, invasive diagnostic and/or therapeutic procedures, time to operation, and place of tumor (rectum vs. rectosigmoid junction). R Statistical Software (Foundation for Statistical Computing, Vienna, Austria) was used for statistical analysis, and *p* < 0.05 was set as significant for the analyses.

## 3. Results

### 3.1. Sex Differences

#### 3.1.1. Non-Elderly Patients

A total of 4213 (38.6%) patients admitted emergently for rectal or rectosigmoid junction malignant neoplasms were females, and 6705 (61.4%) were males with a similar mean age of about 53 years old. Regardless of the sex, most patients were white, funded mostly by private insurance, and were admitted mostly to a teaching hospital (Table 2). Some major comorbidities among the emergency admitted non-elderly patients were AIDS, alcohol abuse, deficiency anemias, and fluid/electrolyte disorders. Males manifested with significantly higher comorbidities of AIDS, alcohol abuse, drug abuse, liver disease, and renal failure, while females showed higher comorbidities of rheumatoid arthritis, depression, and hypothyroidism. Males underwent more invasive diagnostic and/or therapeutic procedures on the gastrointestinal (GI) system and significantly higher rates of digestive system operations. Patients’ characteristics and clinical data are summarized in Table 2.

#### 3.1.2. Elderly Patients

The mean (SD) age of the females, 79 (8), was significantly higher than the mean (SD) age of the males, 76 (8). Regardless of sex, most patients were white and funded mostly by Medicare (Table 2). Major comorbidities among the emergency admitted elderly patients were hypertension, chronic pulmonary disease, uncomplicated diabetes, renal failure, and fluid/electrolyte disorders, among others. Males manifested significantly higher comorbidities with chronic pulmonary disease, alcohol abuse, peripheral vascular disorders, and renal failure, while females showed higher comorbidities of rheumatoid arthritis, depression, hypothyroidism, obesity, and fluid/electrolyte disorders. Males had less time to surgical procedure. Patients’ characteristics and clinical data are summarized in Table 2.

### 3.2. Mortality

#### 3.2.1. Adult Patients

A total of 95.5% of patients survived, and 4.5% died within the immediate hospital stay. The mean (SD) age of the patients who survived was 52.81 (8.57) years: 6416 were males (61.6%), and 4003 were females (38.4%), with a similar mean age. The mean (SD) age of the 42 patients who died during the study period was 53.87 (8.05) years: 288 were males (58.18%), and 207 were females (41.82%), with a similar mean age.

When comparing deceased to survived patients, significant differences were noted in terms of certain morbidities. The patients who died manifested significantly higher rates of the following comorbidities: coagulopathy, liver disease, fluid/electrolyte disorders, pulmonary circulation disorders, and renal failure. The adult deceased patients manifested significantly higher rates of rectosigmoid junction malignancies and lower rates of rectal malignancies in comparison to the adult patients that survived. The deceased patients also showed a significantly higher rate of having undergone invasive diagnostic and/or therapeutic procedures on the GI system and a higher rate of undergoing a digestive system operation. Patients’ characteristics and clinical data are summarized in Table 3.

#### 3.2.2. Elderly Patients

A total of 11,933 patients (94.0%) lived and 763 (6%) died within the immediate hospital stay. The patients who survived included 5650 females (47.4%) and 6283 (52.6%) males. The mean (SD) age of the 355 patients who died during the study period was significantly higher in comparison to the patients who survived, 78.55 (8.00) vs. 77.34 (7.92), respectively. When comparing deceased to survived patients, significant differences were noted in terms of morbidities. The deceased patients manifested with significantly higher rates of comorbidities with pulmonary circulation disorders, fluid/electrolyte disorders, coagulopathy, liver disease, and weight loss. The elderly deceased patients had significantly higher rates of rectosigmoid junction and lower rates of rectal malignancies in comparison to the patients that survived. The deceased patients also had a significantly lower rate of undergoing invasive diagnostic and/or therapeutic procedures on the GI system, a lower rate of digestive system operations, and a longer time to invasive and surgical procedures. Patients’ characteristics and clinical data are summarized in Table 3.

### 3.3. Operation vs. No Operation

#### 3.3.1. Adult Patients

The stratified analysis, based on the surgical procedure status, is presented in Table 4. The mean (SD) age of the patients who had a surgical procedure was significantly higher in comparison to the no surgery group, 53.15 (8.39) years vs. 52.50 (8.74) years, respectively. In both groups, most patients were males. The racial breakdown, by proportion of cases in decreasing order was White, Black, Hispanic, Asian/Pacific Islander, and Native American. Most patients were funded mostly by private insurance and were admitted to urban teaching hospitals. In the group that had a surgical procedure, the rate of comorbidities, such as metastatic cancer and solid tumors, was significantly higher in comparison to the other group. They furthermore manifested with a higher rate of rectosigmoid junction neoplasm and of rectal neoplasm, a higher rate of invasive diagnostic and/or therapeutic procedures on GI, a longer time to invasive diagnostic and/or therapeutic procedure, a longer hospital length of stay (HLOS), and a lower mortality. Patients’ characteristics and clinical data are summarized in Table 4. In total, 44.3% of the surgical procedures on adult patients with this diagnosis were operations on the intestines (ICD-9 codes 45.00–45.03, 45.30–46.99), which included excisions and colostomies. Similarly, 42.8% of the invasive diagnostic and/or therapeutic procedures were performed on the intestines (ICD-9 codes 45.11–45.29).

#### 3.3.2. Elderly Patients

The stratified analysis, based on the surgical procedure status, is also presented in Table 4. Out of 12,701 emergency admitted elderly patients with the primary diagnosis of malignant neoplasm of the rectum and rectosigmoid junction, 7238 (57.0%) had a surgical procedure. The mean (SD) age of the patients who went through a surgical procedure was significantly lower in comparison to the no surgery group, 76.90 (8.39) years vs. 78.09 (8.19) years, respectively. In both groups, most patients were males. The racial breakdown by proportion of cases in decreasing order was White, Black, Hispanic, Asian/Pacific Islander, and Native American. Most patients were funded mostly by Medicare and were admitted to urban teaching hospitals. In the group that had a surgical procedure, the rate of comorbidities, such as metastatic cancer, solid tumors, hypertension, and fluid/electrolyte disorders, was significantly higher in comparison to the other group. They furthermore manifested a higher rate of rectosigmoid junction rectal neoplasm, a higher rate of invasive diagnostic and/or therapeutic procedures on the GI system, a longer HLOS, and significantly lower mortality rates. Patients’ characteristics and clinical data are summarized in Table 4. For elderly patients, 42.5% of the surgical procedures and 47.6% of the invasive diagnostic and/or therapeutic procedures were conducted on the intestine.

### 3.4. Risk Factors of Mortality

The multivariable logistic regression model with backward elimination for mortality was built for the patients that underwent an operation and compared with the model built for the group that did not undergo an operation. The findings are presented in Table 5 and Table 6. Common variables included were age, sex, comorbidities, social factors, lifestyle, and invasive diagnostic and/or therapeutic procedures. Time to surgery was added to the regression model built for patients with operation. HLOS was added to the model built for the group with no operation.

#### 3.4.1. Operated Adult Patients

Table 5 compares mortality in adults and elderly patients that underwent an operation. For adult patients, one day of increased time to operation increased the odds of mortality by 4%. Undergoing invasive diagnostic and/or therapeutic procedures reduced the odds of mortality by 51%. Bacterial infections increased the mortality odds by 2.94-fold. Respiratory diseases similarly elevated the odds by 2.83 times. Patients with a coagulopathy had higher odds of dying by 58%, while those with cardiac disease experienced a 93% increase. Additional risk factors were fluid and electrolyte disorders, liver disease, and neoplasms.

#### 3.4.2. Operated Elderly Patients

Similar risk factors of mortality for both elderly and adult patients included age, time to operation, bacterial infections other than tuberculosis, respiratory diseases, coagulopathy, and fluid and electrolyte disorders. For every additional year of age, patients demonstrated higher mortality odds of 3%. Each day of delay to surgery elevated the odds of death by 5%. Being female offered a protective mortality benefit of 22%. Respiratory infections increased the odds of mortality by 2.61-fold. Patients with a coagulopathy manifested a 48% increase in odds of mortality.

#### 3.4.3. Non-Operated Adult Patients

Table 6 compares the mortality between non-operated adults and elderly patients. Undergoing an invasive diagnostic and/or therapeutic procedure provided an 87% protective mortality benefit for these patients. For liver diseases, patients demonstrated a 95% increase in mortality odds. Having a malignant neoplasm of the rectosigmoid junction, as opposed to that of the rectum, manifested 36% higher odds of mortality. Respiratory diseases elevated the mortality odds by 2.06-fold, while liver disease increased it by 1.95-fold. Other risk factors included fluid and electrolyte disorders, neoplasms, coagulopathy, neurological diseases, skin disease and trauma, burns, and poisons. HLOS was not a risk factor of mortality in these patients.

#### 3.4.4. Non-Operated Elderly Patients

Common risk factors of elderly and adult patients who had no operation were having a malignant neoplasm of the rectosigmoid joint, respiratory disease, and fluid and electrolyte disorders. In the elderly group, each additional year of age raised the mortality risk by 1%. Those that underwent invasive diagnostic and/or therapeutic procedures demonstrated an 81% decrease in mortality odds. Patients with a cancerous lesion of the rectosigmoid junction, as opposed to the rectum had 51% higher odds of dying. For each additional day in the hospital, their mortality risk increased by 2%.

### 3.5. Possible Causes of Mortality

Appendix A summarizes the secondary diagnoses in adult and elderly patients, comparing those that survived vs. those that passed away in the immediate hospital stay. For both adult and elderly patients, some of the pertinent possible causes of death included bacterial infections not including tuberculosis, diabetes, hypertension, coagulopathy, and cardiac disease.

#### 3.5.1. Adult Patients

The possible causes of death in this patient population included nonbacterial infections, diabetes or chronic diabetes complications, anemia and/or hemorrhage, respiratory diseases, and skin diseases, among many others (Appendix A).

#### 3.5.2. Elderly Patients

Some of the possible causes of death in this population included liver disease, disease of the digestive tract, neurological disease, diseases of the musculoskeletal system, tobacco use, platelet and white blood cell disorders, and many others (Appendix A).

## 4. Discussion

The primary aim of this study was to evaluate associations between demographics, socioeconomic status, comorbidities, time to operation, surgical procedure status, postoperative HLOS, and mortality in emergency patients with primary diagnosis of malignant neoplasm of the rectum and rectosigmoid junction. Emergent colorectal procedures are associated with significant morbidity and mortality [11]. Risk factors for adverse outcomes following emergency surgery for colon cancer complications are still debated. Our results demonstrated that in emergency admitted patients with the primary diagnosis of malignant neoplasm of the rectum and rectosigmoid junction, time to surgery, hospital length of stay, age, place of malignancy, and several comorbidities were the main risk factors of mortality, whereas invasive diagnostic and/or therapeutic procedures served as a protective factor. Some factors were repeatedly found to be the predictors of mortality, including emergency surgery, age, patient health status [12,13], grade of the malignant tumor, number and location of metastases, and resection margin [14].

### 4.1. The Impact of HLOS on the Mortality Risk

Our analysis showed in Table 6 that in elderly patients that did not undergo an operation, each additional day in the hospital increased the odds of mortality by 2%. In support of our results, Van Vliet et al. have shown that as HLOS has become significantly shorter for more patients, older patients are less often exposed to the hazards of longer hospital admissions, such as decline in mobility, activities in daily living, and mortality [15]. Better functional outcomes and lower mortality are associated with short admissions, which suggests an advantage of the decrease in HLOS. Enhanced recovery after surgery programs (ERAS) have been introduced with aims of improving patient care, reducing complication rates, and shortening hospital stay following colorectal surgery [16,17]. ERAS have been shown to sustain their effects on emergency surgery patients as well [18,19] with no impact on reducing mortality rates [20,21,22]. A valuable study to analyze the impact of ERAS on HLOS for emergency surgery cases should aim to test the model of Balvardi et al. and compare the patient and hospital characteristics that are associated with discrepancies between HLOS and “time-to-readiness for discharge” measures, as emergency surgery is not included in their study [23]. The crucial role of preoperative assessment, which is omitted in emergency cases and therefore shortens HLOS, including obtaining the necessary clearances and anesthesia evaluation, is well-established [24]. An inability to move the patient to a stable preoperative course, by preventing the preoperative assessment, can be a factor contributing to an increased mortality risk as well, as it could have importance in terms of an underlying propensity for decompensation.

### 4.2. The Impact of Delay in Operation on the Mortality Risk

Our results demonstrated in Table 5 that in emergently admitted patients with the primary diagnosis of a malignant neoplasm of the rectum or rectosigmoid junction that have undergone an operation, time to surgery was among the main risk factors of mortality. Each day delay to the first surgery elevated the odds of mortality by 4% in adult and 5% in elderly patients. Minimizing the delay to definitive operative care may improve outcomes [25]. At present, there are insufficient data regarding the impact of delay in emergency operation and prolonged stay in the hospital on the mortality of emergency admitted patients with the primary diagnosis of malignant neoplasm of the rectum and rectosigmoid junction. Grass et al. assessed the impact of delay from diagnosis to curative surgery on survival in patients with non-metastatic colon cancer and observed that the adjusted hazard ratio for mortality increased with delay times of longer than 30 days, to become significant after a delay of 40 days [26]. In prior NIS database studies, emergently admitted patients with ventral hernia, chronic duodenal ulcers, or hemorrhoids with previous surgery exhibited longer time to operation among the main risk factors for mortality [27,28,29,30].

### 4.3. The Impact of Age on the Mortality Risk

Our findings demonstrated that age was a risk factor of mortality in elderly patients regardless of operation status. Overall, the mortality rate was 4.5% in adult and 6% in elderly patients in the current study, and for every one year that the elderly patient became older, the odds of mortality increased by 1–3%. In the general population, healthy adults from the ages of 18 to 64 have an additional chance of death ranging from 0.02% to 0.3% per year as they grow older. It has previously been shown that young to middle-aged adults have a better overall survival of colorectal cancer as compared to both patients over sixty-five and patients under twenty-one [31]. This difference in survival might partly be explained by the burden of more advanced stage cancers in younger patients and higher postoperative complication risks in older adults because of comorbidities. Although age could be a major player in the algorithm for determining the appropriateness of surgery, the emergency nature of these procedures combined with a poor response to alternative treatments makes surgery an often-imperative risk to take. This finding suggests an opportunity for patient care optimization in patients undergoing emergency surgery for the complications of colon cancer.

A mortality rate at least two to three times higher among the elderly than among younger patients has been repeatedly reported in various populations [32]. In the Netherlands, advanced age and acute operation are by far the most important factors related to operative mortality after colorectal resection [33,34].

### 4.4. The Impact of Sex on the Mortality Risk

The current study showed that being of the female sex is a risk factor in elderly patients that do not undergo an operation but is protective in elderly patients that do undergo an operation. The impact of sex on colorectal cancer incidence is well established. In support of our results, previous studies showed that at all ages, women are less likely to develop colorectal cancers than men [35,36]. In the Women’s Health Initiative trial, post-menopausal estrogen and progestin use was associated with a 40% decrease in colorectal cancer, indicating a role in colorectal cancer carcinogenesis and tumor progression [37]. These hormones are found to also protect pre-menopausal women, as oral contraceptive use reduces the risk of developing colorectal cancers by approximately 20% [38]. We demonstrated that female sex is a predictor for improved survival, and as supported by several studies, male sex adversely affects survival following surgery for colorectal cancer [39,40].

Siegal et al. demonstrated that mortality rate is 30–40% higher in men than in women, although this difference varies by age [3]. A sedentary lifestyle and associated obesity further increase the risk of colon cancer mostly in men [41,42]. Interestingly, and in contrast to previous studies [43,44], our results demonstrate that females sustain their advantage at all ages until 85+, which suggests that the estrogen-related protection is not the only factor associated with their survival.

### 4.5. Rectal vs. Rectosigmoid Malignancy

We have shown in Table 6 that the patients with rectosigmoid junction cancer had 36 and 51 percent higher rates of mortality than those with rectal cancer in adult and elderly patients, respectively. The International Classification of Diseases has recognized the rectosigmoid junction as a unique element and a transition zone separating the sigmoid colon and rectum for further diversification in management and outcomes [45]. The American Joint Committee on Cancer (AJCC) staging system and the Surveillance, Epidemiology, and End Results Program (SEER) database have also recognized the rectosigmoid as a distinct segment; however, currently, cancer of the rectosigmoid junction is still being treated as colon cancer. Previous studies have shown a poor outcome of cancers of the rectosigmoid segment [46,47], which are believed to be associated with different patterns of lymphatic spread with earlier or more frequent metastases to pararectal nodes [48] and therefore likely appear more advanced in presentation. It might be that a wrong-site allocation by endoscopic or radiological test results in the miscoding of upper rectal or distal sigmoid as rectosigmoid cancers [49]. Since the therapeutic options are significantly different when addressing malignancies of the rectum and rectosigmoid, there is an effort to form a consistent definition and to reach a consensus [50,51]. Additional analysis is required to discover the key cause of these fundamental differences between the rectum and the rectosigmoid malignancy.

### 4.6. Surgical Status

Our results in Table 5 and Table 6 showed that the impact of the surgery status on mortality rate was multifactorial and inclusive. In support, the scientific community did not achieve consensus as well. Antony et al. have concluded after risk adjustment that urgent surgery in colon cancer has no impact on survival [52]. On the other hand, Ramos et al. claimed that there was a high mortality rate and a low survival rate in colorectal cancer patients operated on urgently [53]. Smothers et al. claimed that emergency surgery has a strong negative influence on immediate surgical morbidity and mortality; however, other coexistent factors, such as advanced disease, the age of the patient, and medical comorbid conditions, may also influence these outcomes [54]. Goldstone et al. added that operative approach and surgeon training have a substantial impact on outcomes following urgent/emergent colon surgery [55]. Postoperative mortality was two-fold greater when non-colorectal surgeons performed primary anastomosis vs. the Hartmann procedure. Further research is required to evaluate the impact of neoplasm characteristics, timing, and surgical approach on mortality.

### 4.7. Comorbidities and Possible Causes of Death

As expected, comorbid conditions had a great impact on postoperative outcomes and possible mortality, as seen in Table 5 and Table 6 and Appendix A. Common comorbidities between both adult and elderly patients shown as risk factors included coagulopathy, hypertension, liver disease, fluid/electrolyte disorders, metastatic cancer, and pulmonary circulation disorders, among others. In support of our results, studies showed that in patients undergoing colorectal surgery, emergent surgery, liver disease, total colectomy, age older than 65 years, chronic renal failure, and malignant tumor were the major risk factors for in-hospital mortality [56]. Coagulopathy, pulmonary circulatory disorders, liver disorders, renal failure, fluid and electrolyte abnormalities, solid tumors, metastasis, weight loss, AIDS, and alcohol abuse were found to be associated with increased mortality rates. Such mortality rates in patients with coagulopathy may be explained by the increased risk and severity of malignant neoplasm of the rectum and rectosigmoid junction cancer bleeding [57,58]. Moreover, coagulopathies add to the perioperative risk in case surgical hemostasis is indicated [59]. The significantly increased mortality seen in patients with fluid/electrolyte disorders, as presented in Table 5 and Table 6 and Appendix A, is consistent with previous studies, which have highlighted the significant prevalence of hyponatremia in patients with colorectal cancer and the severity-dependent increase in mortality in these patients [60,61]. Hypernatremia was found to be a comorbid condition that was strongly predictive of perioperative death [62]. Congestive heart failure is intimately associated with electrolyte balance, perfusion status, and overall robustness to the trauma of both the malignancy as well as the surgical intervention. Therefore, it has been associated with increased rates of surgical complications and death [62].

### 4.8. The Impact of Invasive Diagnostic and Therapeutic Procedures on Mortality

Our study shows that an invasive diagnostic served as a protective factor in Table 5 and Table 6. In support of these data, Lasisi and Rex have shown improved protection against proximal colon cancer by cecal intubation in screening examinations [62]. Additional studies proved that colonoscopy was strongly associated with reduced odds of both distal and proximal colorectal cancer [63], offered onco-protective effects in both the left and right colon, and confirmed the positive impact on survival in both locations [64]. Colonoscopy was found to be the most effective strategy for detection of precancerous lesions, including large conventional adenomas and large serrated lesions [65]. These findings confirm the need for the continued improvement of invasive diagnostic and therapeutic procedures, their effectiveness, and obligatory quality assessment to optimize the diagnostic yield and its protective factor.

### 4.9. Strengths of the Study

The combination approach of the logistic regression model and the thorough NIS database was the main strength of the current study. This database contains data of patients from multiple states and includes health-based, administrative, and population-based data in a uniform format. The purpose of these data is to improve healthcare through research by analyzing broad combinations of disease conditions, treatments, and outcomes in a large sample size over a ten-year span. Previous studies in the literature focused on smaller subsets of the population over a smaller geographical location in a shorter period of time. This study serves to fill a part of the gap in the literature on the demographics of adult and elderly patients suffering from a rectal or rectosigmoid malignancy as well as their individual hospital course, disease management, and the eventual result of their care.

### 4.10. Limitations of the Study

As with any retrospective cohort analysis, there is a level of inherent limitation to the data; hence, it is important to interpret the data with that in mind. Due to the fact that it is a retrospective study using an administrative dataset, there are variables that cannot be obtained, which would greatly help to further contextualize the results. Given that the principal goal of the study was to identify factors that influence in-hospital mortality in emergency surgery of colon cancer, it would be very helpful to add an additional element of analysis of the causes of death among patients to stratify the associations noted by cause. Another such contextualizing aspect that is missing, given the analysis of patient disposition, is an assessment of functional ability prior to emergency surgery, which would be beneficial. By identifying the level of care required by the patient, such as home health or skilled nursing facility, there would be a more nuanced understanding of whether the disposition is a return to the status quo or an escalation in care status needed by the patient. Additionally, given that comorbidities were among the most influential factors that increased mortality in both associations, it would be helpful to have an understanding of how multiple comorbidities can potentially interact, or simply co-exist, to influence survival following the trauma of abdominal surgery, due to synergistic effects on physical stress. Along those lines, there is potential for a circular influence of the predictors of mortality on outcome. For example, the hospital length of stay can predict the risk of mortality. However, the stay in the hospital will be shorter if the patient dies. Similarly, if a patient undergoes an emergency surgery, that in itself signifies a worse prognosis. Another potential limitation is that the use of a backwards elimination without validation on a separate dataset can potentially overestimate any associations.

The lack of consensus over the transition point for the end of the sigmoid and beginning of the rectum is a problem for the colorectal multidisciplinary team and could have impacted our results [46]. Without a reliable definition of the rectum, rectosigmoid and rectal cancers will be classified inconsistently. As the treatment strategies for rectosigmoid and rectal cancers are radically different, incorrect tumor localization has a substantial impact on patient management, leading to under or over treatment [66]. If a rectal tumor is misclassified as a rectosigmoid tumor, the patient could be inadequately staged and not considered for preoperative downstaging radiation, potentially decreasing their chance of undergoing a complete resection and worsening their survival [67]. Although this is a situational clinical limitation, it could have had a distorting impact on our results. An additional limitation of this study is a lack of the specification of the tumor origin and size, shape of the tumor, perforation and obstruction status, degree of differentiation, proper staging and localization using CT/MRI [68,69] and venous invasion, number of metastases, relevant GI neoplasm family history, operative approach and surgeon training, and whether the diagnosis was made using radiological markings, endoscopic measurements, or anatomical landmarks [70,71]. Further research on the complexity of cases and other modifiable patient factors that could influence patient discharge is necessary.

## 5. Conclusions

In conclusion, an increased time to first surgical procedure, bacterial infections, coagulopathy, cardiac disease, and respiratory disease are risk factors for in-hospital mortality in both adult and elderly patients undergoing emergency surgery for malignant neoplasm of the rectum and rectosigmoid junction, among others. Malignant neoplasm of the rectosigmoid junction as opposed to the rectum or other neoplasms, fluid and electrolyte disorders, and respiratory disease were risk factors, among others, for in-hospital mortality in adult and elderly patients that did not undergo an operation. Liver disease, coagulopathy, and cardiac disease were risk factors for mortality in non-operated adult patients, while age, hospital length of stay, and female sex were risk factors for non-operated elderly patients.

## Figures and Tables

**Table 1 ijerph-19-09203-t001:** Procedures of emergency admitted patients with the primary diagnosis of rectum or rectosigmoid junction.

**Operations on the Digestive System (ICD 9)**
Operations on Esophagus (42.01–42.19, 42.31–42.99)
Operations on Stomach (43.0–44.03, 44.21–44.99)
Operations on Intestine (45.00–45.03, 45.30–46.99)
Operations on Appendix (47.01–47.99)
Operations on Rectum, Rectosigmoid, and Perirectal Tissue (48.0–48.1, 48.31–48.99)
Operations on Anus (49.01–49.12, 49.31–49.99)
Operations on Liver (50.0, 50.21–50.99)
Operations on Gallbladder and Biliary Tract (51.01–51.04, 51.21–51.99)
Operations on Pancreas (52.01–52.09, 52.21–52.99)
Operations on Hernia (53.00–53.9)
Operations on Other Operations on Abdominal Region (54.0–54.19, 54.3–54.99)
**Invasive Diagnostic and Therapeutic Procedures on the Digestive System (ICD 9)**
Invasive Diagnostic and/or Therapeutic Procedure on Esophagus (42.21–42.29)
Invasive Diagnostic and/or Therapeutic Procedure on Stomach (44.11–44.19)
Invasive Diagnostic and/or Therapeutic Procedure on Intestine (45.11–45.29)
Invasive Diagnostic and/or Therapeutic Procedure on Rectum, Rectosigmoid, and Perirectal Tissue (48.21–48.29)
Invasive Diagnostic and/or Therapeutic Procedure on Anus (49.21–49.29)
Invasive Diagnostic and/or Therapeutic Procedure on Liver (50.11–50.19)
Invasive Diagnostic and/or Therapeutic Procedure on Gallbladder and Biliary Tract (51.10–51.19)
Invasive Diagnostic and/or Therapeutic Procedure on Pancreas (52.11–52.19)
Invasive Diagnostic and/or Therapeutic Procedure on Other Operations on Abdominal Region (54.21–54.29)

**Table 2 ijerph-19-09203-t002:** Characteristics of emergency admitted patients with the primary diagnosis of malignant neoplasm of rectum or rectosigmoid junction. Data (NIS 2005–2014) were stratified according to sex categories. * *p* < 0.05.

Patients’ Characteristics		Adult (18–64), N (%)		Elderly (65+), N (%)	
		Male	Female	Male	Female
All Cases		6705 (61.4%)	4213 (38.6%)	6708 (52.8%)	5988 (47.2%)
Race	White	3506 (59.9%) *	2298 (63.2%) *	4304 (74.9%) *	3830 (74.8%) *
	Black	1040 (17.8%) *	624 (17.2%) *	574 (10.0%) *	630 (12.3%) *
	Hispanic	805 (13.7%) *	431 (11.9%) *	490 (8.5%) *	331 (6.5%) *
	Asian/Pacific Islander	269 (4.6%) *	142 (3.9%) *	203 (3.5%) *	163 (3.2%) *
	Native American	51 (0.9%) *	27 (0.7%) *	34 (0.6%) *	31 (0.6%) *
	Other	185 (3.2%) *	112 (3.1%) *	145 (2.5%) *	133 (2.6%) *
Income Quartile	Quartile 1	2144 (33.1%)	1319 (32.1%)	1943 (29.6%)	1645 (28.0%)
	Quartile 2	1717 (26.5%)	1041 (25.3%)	1775 (27.0%)	1549 (26.4%)
	Quartile 3	1466 (22.6%)	957 (23.3%)	1525 (23.2%)	1384 (23.6%)
	Quartile 4	1152 (17.8%)	798 (19.4%)	1326 (20.2%)	1288 (22.0%)
Insurance	Private Insurance	2774 (41.5%) *	1923 (45.8%) *	639 (9.5%) *	430 (7.2%) *
	Medicare	763 (11.4%) *	426 (10.1%) *	5800 (86.6%) *	5313 (88.8%) *
	Medicaid	1781 (26.6%) *	1095 (26.1%) *	111 (1.7%) *	127 (2.1%) *
	Self-Pay	882 (13.2%) *	462 (11.0%) *	50 (0.7%) *	55 (0.9%) *
	No Charge	99 (1.5%) *	58 (1.4%) *	5 (0.1%) *	5 (0.1%) *
	Other	384 (5.7%) *	235 (5.6%) *	90 (1.3%) *	51 (0.9%) *
Hospital Location	Rural	673 (10.0%)	399 (9.5%)	905 (13.5%)	778 (13.0%)
	Urban: Non-Teaching	2471 (36.9%)	1493 (35.4%)	2834 (42.2%)	2616 (43.7%)
	Urban: Teaching	3561 (53.1%)	2321 (55.1%)	2969 (44.3%)	2594 (43.3%)
Comorbidities	AIDS	92 (1.4%) *	25 (0.6%) *	4 (0.1%)	0 (0%)
	Alcohol Abuse	580 (8.7%) *	92 (2.2%) *	286 (4.3%) *	55 (0.9%) *
	Deficiency Anemias	1914 (28.5%) *	1279 (30.4%) *	2000 (29.8%)	1802 (30.1%)
	Rheumatoid Arthritis	18 (0.3%) *	68 (1.6%) *	42 (0.6%) *	144 (2.4%) *
	Chronic Blood Loss	635 (9.5%) *	340 (8.1%) *	827 (12.3%)	793 (13.2%)
	Congestive Heart Failure	187 (2.8%)	124 (2.9%)	875 (13.0%)	806 (13.5%)
	Chronic Pulmonary Disease	629 (9.4%) *	452 (10.7%) *	1367 (20.4%) *	1021 (17.1%) *
	Coagulopathy	280 (4.2%)	171 (4.1%)	339 (5.1%) *	240 (4.0%) *
	Depression	381 (5.7%) *	457 (10.8%) *	389 (5.8%) *	476 (7.9%) *
	Diabetes Uncomplicated	896 (13.4%)	555 (13.2%)	1427 (21.3%) *	1149 (19.2%) *
	Diabetes, Chronic Complications	125 (1.9%)	59 (1.4%)	190 (2.8%)	155 (2.6%)
	Drug Abuse	269 (4.0%) *	87 (2.1%) *	24 (0.4%)	13 (0.2%)
	Hypertension	2333 (34.8%)	1439 (34.2%)	3841 (57.3%) *	3691 (61.6%) *
	Hypothyroidism	129 (1.9%) *	289 (6.9%) *	359 (5.4%) *	930 (15.5%) *
	Liver Disease	331 (4.9%) *	120 (2.8%) *	139 (2.1%)	99 (1.7%)
	Lymphoma	17 (0.3%)	10 (0.2%)	31 (0.5%)	25 (0.4%)
	Fluid/Electrolyte Disorders	1958 (29.2%) *	1385 (32.9%) *	2262 (33.7%)	2402 (40.1%)
	Metastatic Cancer	1552 (23.1%)	967 (23.0%)	1214 (18.1%)	1051 (17.6%)
	Other Neurological Disorders	210 (3.1%) *	162 (3.8%) *	435 (6.5%) *	477 (8.0%) *
	Obesity	374 (5.6%) *	346 (8.2%) *	252 (3.8%) *	327 (5.5%) *
	Paralysis	69 (1.0%)	43 (1.0%)	138 (2.1%)	106 (1.8%)
	Peripheral Vascular Disorders	143 (2.1%) *	56 (1.3%) *	533 (7.9%) *	357 (6.0%) *
	Psychoses	227 (3.4%) *	179 (4.2%) *	160 (2.4%)	168 (2.8%)
	Pulmonary Circulation Disorder	80 (1.2%)	53 (1.3%)	188 (2.8%)	190 (3.2%)
	Renal Failure	293 (4.4%) *	109 (2.6%) *	851 (12.7%) *	537 (9.0%) *
	Solid Tumor	74 (1.1%)	49 (1.2%)	122 (1.8%)	83 (1.4%)
	Peptic Ulcer	1 (0.0%)	2 (0.0%)	10 (0.1%)	6 (0.1%)
	Valvular Disease	85 (1.3%)	65 (1.5%)	455 (6.8%)	432 (7.2%)
	Weight Loss	1095 (16.3%) *	595 (14.1%) *	1200 (17.9%)	993 (16.6%)
Neoplasm Location	Rectosigmoid Junction	2870 (42.8%) *	1886 (44.8%) *	2703 (40.3%) *	2533 (42.3%) *
	Rectum	3835 (57.2%) *	2327 (55.2%) *	4005 (59.7%) *	3455 (57.7%) *
Invasive Diagnostic and/or Therapeutic Procedures on GI		3477 (51.9%) *	1997 (47.4%) *	3699 (55.1%)	3382 (56.5%)
GI System Operation		3746 (55.9%) *	2259 (53.6%) *	3930 (58.6%) *	3304 (55.2%) *
Deceased		288 (4.3%)	207 (4.9%)	428 (6.4%)	335 (5.6%)
		Mean (SD)	Mean (SD)	Mean (SD)	Mean (SD)
Age, Years		53.03 (8.41) *	52.59 (8.78) *	76.24 (7.56) *	78.72 (8.12) *
Time to Invasive Diagnostic and/or Therapeutic Procedure, Days		2.03 (2.94)	2.18 (3.16)	2.26 (2.90)	2.37 (2.65)
Time to Surgical Procedure, Days		2.51 (3.95)	2.55 (3.56)	2.92 (3.96) *	3.28 (3.76) *
Hospital Length of Stay, Days		8.67 (9.09)	8.56 (10.05)	9.44 (9.06)	9.18 (7.73)
Total Charges, USD		64,420 (99,057)	60,456 (81,658)	66,654 (91,353)	60,625 (69,078)

**Table 3 ijerph-19-09203-t003:** Characteristics of emergency admitted patients with the primary diagnosis of malignant neoplasm of rectum or rectosigmoid junction. Data (NIS 2005–2014) were stratified according to outcome categories. * *p* < 0.05.

Patients’ Characteristics		Adult (18–64), N (%)		Elderly (65+), N (%)	
		Survived	Deceased	Survived	Deceased
All Cases		10,419 (95.5%)	495 (4.5%)	11,933 (94.0%)	763 (6.0%)
Sex, Female		4003 (38.4%)	207 (41.8%)	5650 (47.4%)	335 (43.9%)
Race	White	5517 (60.9%)	284 (66.7%)	7622 (74.6%)	508 (78.3%)
	Black	1598 (17.6%)	65 (15.3%)	1146 (11.2%)	58 (8.9%)
	Hispanic	1194 (13.2%)	42 (9.9%)	778 (7.6%)	42 (6.5%)
	Asian/Pacific Islander	389 (4.3%)	21 (4.9%)	350 (3.4%)	16 (2.5%)
	Native American	77 (0.8%)	1 (0.2%)	59 (0.6%)	6 (0.9%)
	Other	284 (3.1%)	13 (3.1%)	259 (2.5%)	19 (2.9%)
Income Quartile	Quartile 1	3321 (32.9%)	142 (29.3%)	3372 (28.9%)	220 (29.4%)
	Quartile 2	2626 (26.0%)	132 (27.2%)	3120 (26.7%)	202 (27.0%)
	Quartile 3	2316 (22.9%)	106 (21.9%)	2749 (23.5%)	160 (21.4%)
	Quartile 4	1844 (18.2%)	105 (21.6%)	2446 (20.9%)	167 (22.3%)
Insurance	Private Insurance	4541 42.8(%)	242 (49.6%)	967 (8.1%) *	102 (13.4%) *
	Medicare	1135 (10.9%)	54 (11.1%)	10,501 (88.1%) *	612 (80.4%) *
	Medicaid	2776 (26.7%)	103 (21.1%)	226 (1.9%) *	12 (1.6%) *
	Self-Pay	1290 (12.4%)	53 (10.9%)	96 (0.8%) *	9 (1.2%) *
	No Charge	153 (1.5%)	4 (0.8%)	9 (0.1%) *	1 (0.1%) *
	Other	585 (5.6%)	32 (6.6%)	116 (1.0%) *	25 (3.3%) *
Hospital Location	Rural	1002 (9.6%)	70 (14.1%)	1560 (13.1%)	123 (16.1%)
	Urban: Non-Teaching	3799 (36.5%)	161 (32.5%)	5128 (43.0%)	320 (41.9%)
	Urban: Teaching	5618 (53.9%)	264 (53.3%)	5245 (44.0%)	320 (41.9%)
Comorbidities	AIDS	113 (1.1%)	4 (0.8%)	3 (0.0%)	1 (0.1%)
	Alcohol Abuse	652 (6.3%)	20 (4.0%)	329 (2.8%)	12 (1.6%)
	Deficiency Anemias	3080 (29.6%)	111 (22.4%)	3622 (30.4%) *	179 (23.5%) *
	Rheumatoid Arthritis	84 (0.8%)	2 (0.4%)	180 (1.5%)	6 (0.8%)
	Chronic Blood Loss	955 (9.2%) *	20 (4.0%) *	1564 (13.1%) *	56 (7.3%) *
	Congestive Heart Failure	288 (2.8%)	23 (4.6%)	1546 (13.0%) *	135 (17.7%) *
	Chronic Pulmonary Disease	1037 (10.0%)	42 (8.5%)	2237 (18.7%)	151 (19.8%)
	Coagulopathy	372 (3.6%) *	79 (16.0%) *	490 (4.1%) *	88 (11.5%) *
	Depression	823 (7.9%) *	16 (3.2%) *	815 (6.8%)	49 (6.4%)
	Diabetes, Uncomplicated	1403 (13.5%)	46 (9.3%)	2465 (20.7%) *	111 (14.5%) *
	Diabetes, Chronic Complications	172 (1.7%)	12 (2.4%)	334 (2.8%)	11 (1.4%)
	Drug Abuse	350 (3.4%)	6 (1.2%)	35 (0.3%)	2 (0.3%)
	Hypertension	3645 (35.0%) *	125 (25.3%) *	7181 (60.2%) *	351 (46.0%) *
	Hypothyroidism	400 (3.8%)	18 (3.6%)	1230 (10.3%)	59 (7.7%)
	Liver Disease	395 (3.8%) *	56 (11.3%) *	208 (1.7%) *	30 (3.9%) *
	Lymphoma	25 (0.2%)	2 (0.4%)	51 (0.4%)	5 (0.7%)
	Fluid/Electrolyte Disorders	3079 (29.6%) *	263 (53.1%) *	4268 (35.8%) *	392 (51.4%) *
	Metastatic Cancer	2434 (23.4%) *	81 (16.4%) *	2120 (17.8%)	145 (19.0%)
	Other Neurological Disorders	339 (3.3%) *	33 (6.7%) *	844 (7.1%)	68 (8.9%)
	Obesity	689 (6.6%)	31 (6.3%)	564 (4.7%) *	15 (2.0%) *
	Paralysis	104 (1.0%)	7 (1.4%)	230 (1.9%)	14 (1.8%)
	Peripheral Vascular Disorders	187 (1.8%)	12 (2.4%)	832 (7.0%)	58 (7.6%)
	Psychoses	389 (3.7%)	17 (3.4%)	298 (2.5%)	30 (3.9%)
	Pulmonary Circulation Disorder	115 (1.1%) *	18 (3.6%) *	336 (2.8%) *	42 (5.5%) *
	Renal Failure	369 (3.5%) *	33 (6.7%) *	1281 (10.7%)	106 (13.9%)
	Solid Tumor	119 (1.1%)	3 (0.6%)	196 (1.6%)	9 (1.2%)
	Peptic Ulcer	3 (0.0%)	0 (0%)	16 (0.1%)	0 (0%)
	Valvular Disease	146 (1.4%)	4 (0.8%)	846 (7.1%)	41 (5.4%)
	Weight Loss	1593 (15.3%)	97 (19.6%)	1996 (16.7%) *	194 (25.4%) *
Neoplasm Location	Rectosigmoid Junction	4480 (43.0%) *	273 (55.2%) *	4851 (40.7%) *	382 (50.1%) *
	Rectum	5939 (57.0%) *	222 (44.8%) *	7082 (59.3%) *	381 (49.9%) *
Invasive Diagnostic and/or Therapeutic Procedures on GI		5373 (51.6%) *	99 (20.0%) *	6791 (56.9%) *	288 (37.7%) *
GI System Operation		5847 (56.1%) *	154 (31.1%) *	6855 (57.4%) *	379 (49.7%) *
		Mean (SD)	Mean (SD)	Mean (SD)	Mean (SD)
Age, Years		52.81 (8.57)	53.87 (8.05)	77.34 (7.92) *	78.55 (8.00) *
Time to Invasive Diagnostic and/or Therapeutic Procedure, Days		2.04 (2.83) *	4.25 (7.81) *	2.26 (2.61) *	3.44 (5.37) *
Time to First Surgical Procedure, Days		2.47 (3.58)	4.49 (8.57)	2.97 (3.69) *	5.04 (6.04) *
Hospital Length of Stay, Days		8.52 (8.88)	10.89 (17.69)	9.18 (8.21)	11.47 (11.44)
Total Charges, USD		62,082 (88,996) *	80,183 (151,679) *	62,206 (76,350)	88,847 (138,226)

**Table 4 ijerph-19-09203-t004:** Characteristics of emergency admitted patients with the primary diagnosis of malignant neoplasm of rectum or rectosigmoid junction. Data (NIS 2005–2014) were stratified according to surgery status. * *p* < 0.05.

Patients’ Characteristics		Adult (18–64), N (%)		Elderly (65+), N (%)	
		No Operation	Operation	No Operation	Operation
All Cases		4915 (45.0%)	6007 (55.0%)	5463 (43.0%)	7238 (57.0%)
Sex, Female		1954 (39.8%)	2259 (37.6%)	2684 (49.1%) *	3304 (45.7%) *
Race	White	2448 (57.0%) *	3357 (64.6%) *	3422 (72.7%) *	4712 (76.5%) *
	Black	882 (20.6%) *	782 (15.0%) *	624 (13.3%) *	580 (9.4%) *
	Hispanic	607 (14.1%) *	629 (12.1%) *	351 (7.5%) *	470 (7.6%) *
	Asian/Pacific Islander	186 (4.3%) *	225 (4.3%) *	155 (3.3%) *	211 (3.4%) *
	Native American	31 (0.7%) *	47 (0.9%) *	24 (0.5%) *	41 (0.7%) *
	Other	137 (3.2%) *	160 (3.1%) *	131 (2.8%) *	147 (2.4%) *
Income Quartile	Quartile 1	1585 (33.4%)	1881 (32.1%)	1545 (28.9%)	2047 (28.8%)
	Quartile 2	1210 (25.5%)	1549 (26.4%)	1355 (25.4%)	1969 (27.7%)
	Quartile 3	1072 (22.6%)	1351 (23.1%)	1248 (23.4%)	1661 (23.4%)
	Quartile 4	874 (18.4%)	1076 (18.4%)	1194 (22.4%)	1421 (20.0%)
Insurance	Private Insurance	1900 (38.8%) *	2797 (46.7%) *	498 (9.1%) *	571 (7.9%) *
	Medicare	616 (12.6%) *	573 (9.6%) *	4695 (86.1%) *	6423 (88.9%) *
	Medicaid	1420 (29.0%) *	1460 (24.4%) *	121 (2.2%) *	117 (1.6%) *
	Self-Pay	610 (12.5%) *	734 (12.3%) *	51 (0.9%) *	54 (0.7%) *
	No Charge	77 (1.6%) *	80 (1.3%) *	3 (0.1%) *	7 (0.1%) *
	Other	275 (5.6%) *	344 (5.7%) *	85 (1.6%) *	56 (0.8%) *
Hospital Location	Rural	455 (9.3%)	617 (10.3%)	704 (12.9%)	979 (13.5%)
	Urban: Non-Teaching	1730 (35.2%)	2234 (37.2%)	2297 (42.0%)	3154 (43.6%)
	Urban: Teaching	2730 (55.5%)	3156 (52.5%)	2462 (45.1%)	3105 (42.9%)
Comorbidites	AIDS	83 (1.7%) *	34 (0.6%) *	2 (0.0%)	2 (0.0%)
	Alcohol Abuse	276 (5.6%)	396 (6.6%)	155 (2.8%)	186 (2.6%)
	Deficiency Anemias	1550 (31.5%) *	1643 (27.4%) *	1724 (31.6%)	2078 (28.7%)
	Rheumatoid Arthritis	46 (0.9%)	40 (0.7%)	80 (1.5%)	106 (1.5%)
	Chronic Blood Loss	496 (10.1%) *	479 (8.0%) *	786 (14.4%) *	834 (11.5%) *
	Congestive Heart Failure	133 (2.7%)	178 (3.0%)	674 (12.3%)	1007 (13.9%)
	Chronic Pulmonary Disease	424 (8.6%) *	657 (10.9%) *	914 (16.7%) *	1474 (20.4%) *
	Coagulopathy	213 (4.3%)	238 (4.0%)	224 (4.1%)	355 (4.9%)
	Depression	387 (7.9%)	452 (7.5%)	432 (7.9%) *	433 (6.0%) *
	Diabetes, Uncomplicated	644 (13.1%)	807 (13.4%)	1096 (20.1%)	1480 (20.4%)
	Diabetes, Chronic Complications	83 (1.7%)	101 (1.7%)	141 (2.6%)	204 (2.8%)
	Drug Abuse	177 (3.6%)	179 (3.0%)	21 (0.4%)	16 (0.2%)
	Hypertension	1635 (33.3%)	2137 (35.6%)	3104 (56.8%) *	4429 (61.2%) *
	Hypothyroidism	201 (4.1%)	217 (3.6%)	567 (10.4%)	722 (10.0%)
	Liver Disease	193 (3.9%)	258 (4.3%)	107 (2.0%)	131 (1.8%)
	Lymphoma	11 (0.2%)	16 (0.3%)	22 (0.4%)	34 (0.5%)
	Fluid/Electrolyte Disorders	1549 (31.5%)	1794 (29.9%)	1900 (34.8%) *	2765 (38.2%) *
	Metastatic Cancer	204 (4.2%) *	2315 (38.5%) *	123 (2.3%) *	2144 (29.6%) *
	Other Neurological Disorders	189 (3.8%)	183 (3.0%)	427 (7.8%)	485 (6.7%)
	Obesity	253 (5.1%) *	467 (7.8%) *	177 (3.2%) *	402 (5.6%) *
	Paralysis	58 (1.2%)	54 (0.9%)	116 (2.1%)	128 (1.8%)
	Peripheral Vascular Disorders	73 (1.5%)	126 (2.1%)	377 (6.9%)	513 (7.1%)
	Psychoses	194 (3.9%)	212 (3.5%)	143 (2.6%)	185 (2.6%)
	Pulmonary Circulation Disorder	58 (1.2%)	75 (1.2%)	141 (2.6%)	237 (3.3%)
	Renal Failure	203 (4.1%)	199 (3.3%)	631 (11.6%)	757 (10.5%)
	Solid Tumor	6 (0.1%) *	117 (1.9%) *	11 (0.2%) *	194 (2.7%) *
	Peptic Ulcer	2 (0.0%)	1 (0.0%)	9 (0.2%)	7 (0.1%)
	Valvular Disease	57 (1.2%)	93 (1.5%)	301 (5.5%) *	586 (8.1%) *
	Weight Loss	739 (15.0%)	951 (15.8%)	859 (15.7%) *	1334 (18.4%) *
Neoplasm Location	Rectosigmoid Junction	1872 (38.1%) *	2887 (48.1%) *	1876 (34.3%) *	3361 (46.4%) *
	Rectum	3043 (61.9%) *	3120 (51.9%) *	3587 (65.7%) *	3877 (53.6%) *
Invasive Diagnostic and/or Therapeutic Procedures on GI		2226 (45.3%) *	3249 (54.1%) *	2827 (51.7%) *	4254 (58.8%) *
Deceased		341 (6.9%) *	154 (2.6%) *	384 (7.0%) *	379 (5.2%) *
		Mean (SD)	Mean (SD)	Mean (SD)	Mean (SD)
Age, Years		52.50 (8.74) *	53.15 (8.39) *	78.09 (8.19) *	76.90 (7.69) *
Time to Invasive Diagnostic and/or Therapeutic Procedure, Days		2.07 (2.45) *	2.10 (3.38) *	2.28 (2.39)	2.33 (3.03)
Hospital Length of Stay, Days		6.04 (7.30) *	10.75 (10.46) *	6.05 (5.91) *	11.78 (9.23) *
Total Charges, USD		36,314 (47,212) *	84,344 (113,048) *	33,643 (37,915) *	86,498 (96,956) *

**Table 5 ijerph-19-09203-t005:** Backward logistic regression analysis to evaluate the associations between mortality and different risk factors in patients emergently admitted with a primary diagnosis of malignant neoplasm of rectum or rectosigmoid junction and undergoing an operation (NIS 2004–2014). Mortality was the dependent variable.

Patients’ Characteristics	Mortality in Adults Patients with Operation		Mortality in Elderly Patients with Operation	
	N = 6619	R^2^ = 0.249	N = 8167	R^2^ = 0.186
	OR (95% CI)	*p*	OR (95% CI)	*p*
Number of Events	N = 205		N = 485	
Age, Years	Removed		1.03 (1.02, 1.05)	<0.001
Invasive Diagnostic and/or Therapeutic Procedure	0.49 (0.35, 0.67)	<0.001	0.82 (0.66, 1.01)	0.070
Time to First Surgical Operation, Days	1.04 (1.02, 1.07)	<0.001	1.05 (1.03, 1.07)	<0.001
Sex, Female	Removed		0.78 (0.63, 0.96)	0.019
Bacterial Infections (Other than Tuberculosis)	2.94 (1.99, 4.34)	<0.001	2.53 (1.99, 3.21)	<0.001
Respiratory Diseases	2.83 (2.02, 3.98)	<0.001	2.61 (2.08, 3.28)	<0.001
Coagulopathy	1.58 (1.03, 2.43)	0.036	1.48 (1.12, 1.95)	0.006
Cardiac Diseases	1.93 (1.39, 2.70)	<0.001	1.67 (1.33, 2.10)	<0.001
Fluid and Electrolyte Disorders	1.84 (1.31, 2.60)	<0.001	1.82 (1.46, 2.26)	<0.001
Genitourinary System Diseases	1.32 (0.95, 1.84)	0.100	1.29 (1.04, 1.60)	0.020
Trauma, Burns, and Poisons	1.43 (0.99, 2.07)	0.060	1.86 (1.48, 2.33)	<0.001
Liver Diseases	2.51 (1.64, 3.85)	<0.001	Removed Via Stepwise Backward Elimination	
Neoplasms	2.42 (1.67, 3.50)	<0.001		
Neurological Diseases	1.99 (1.34, 2.96)	<0.001		
Neoplasm of Rectosigmoid Junction	Removed Via Stepwise Backward Elimination			
Digestive Diseases other than Liver				
Hypertension				
Anemia and/or Hemorrhage				
Musculoskeletal System and Connective Tissue Diseases				
Tobacco Use				
Psychiatric Diseases				
Endocrine Diseases				
Tuberculosis				
Nonbacterial Infections				
Peripheral Vascular Diseases				
Diabetes				
Alcohol Abuse/Withdrawal/Dependence				
Drug Abuse/Withdrawal/Dependence				
Nutritional/Weight Disorders				
Platelet and White Blood Cell Diseases				
Skin Diseases				
Medications				
Diseases of Oral Cavity, Salivary Glands, and Jaw				
Cerebrovascular Diseases				
Sleep Disorders				
Lack of Physical Evidence				
Inappropriate Diet and Eating Habits				
High Risk Lifestyle Behaviors				
Social Factors				
Body Mass Index				

**Table 6 ijerph-19-09203-t006:** Backward logistic regression analysis to evaluate the associations between mortality and different factors in patients emergently admitted with a primary diagnosis of malignant neoplasm of rectum or rectosigmoid junction and not undergoing an operation (NIS 2004–2014). Mortality was the dependent variable.

Patients’ Characteristics	Mortality in Adult Patients with No Operation		Mortality in Elderly Patients with No Operation	
	N = 5867	R^2^ = 0.210	N = 6873	R^2^ = 0.191
	OR (95% CI)	*p*	OR (95% CI)	*p*
Number of Events	N = 419		N = 521	
Age, Years	Removed		1.01 (1.00, 1.03)	0.032
Invasive Diagnostic and/or Therapeutic Procedures	0.13 (0.09, 0.18)	<0.001	0.19 (0.15, 0.24)	<0.001
Hospital Length of Stay, Days	1.01 (0.99, 1.02)	0.060	1.02 (1.01, 1.03)	0.002
Malignant Neoplasm of Rectosigmoid Junction	1.36 (1.10, 1.68)	0.005	1.51 (1.25, 1.83)	<0.001
Respiratory Diseases	2.06 (1.63, 2.59)	<0.001	1.78 (1.47, 2.16)	<0.001
Fluid and Electrolyte Disorders	1.62 (1.31, 2.01)	<0.001	1.44 (1.19, 1.73)	<0.001
Neoplasms	2.22 (1.68, 2.94)	<0.001	1.25 (1.03, 1.53)	0.028
Liver Diseases	1.95 (1.39, 2.73)	<0.001	Removed Via Stepwise Backward Elimination	
Coagulopathy	1.60 (1.17, 2.18)	0.003		
Cardiac Diseases	1.55 (1.20, 2.00)	<0.001		
Neurological Diseases	1.34 (1.00, 1.80)	0.047		
Skin Diseases	1.42 (1.05, 1.91)	0.023		
Trauma, Burns, and Poisons	1.82 (1.33, 2.47)	<0.001		
Sex, Female	Removed Via Stepwise Backward Elimination		1.89 (1.45, 2.46)	<0.001
Neoplasm of Rectosigmoid Junction			1.47 (1.09, 1.96)	0.011
Bacterial Infections (Other than Tuberculosis)			Removed Via Stepwise Backward Elimination	
Platelet and White Blood Cell Diseases				
Diseases of Oral Cavity, Salivary Glands, and Jaw				
Anemia and/or Hemorrhage				
Digestive Diseases other than Liver				
Tobacco Use				
Hypertension				
Endocrine Diseases				
Musculoskeletal System and Connective Tissue Diseases				
Social Factors				
Medications				
Psychiatric Diseases				
Cerebrovascular Diseases				
Nonbacterial Infections				
Alcohol Abuse/Withdrawal/Dependence				
Peripheral Vascular Diseases				
Diabetes				
Drug Abuse/Withdrawal/Dependence				
Genitourinary System Diseases				
Tuberculosis				
Nutritional/Weight Disorders				
Sleep Disorders				
Lack of Physical Evidence				
Inappropriate Diet and Eating Habits				
High Risk Lifestyle Behaviors				

## Data Availability

Data will be available for verification purposes upon request.

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
