# Peer review of "Adult and Elderly Risk Factors of Mortality in 23,614 Emergently Admitted Patients with Rectal or Rectosigmoid Junction Malignancy"

_ijerph, 2022, doi:10.3390/ijerph19159203_

Round 1

Reviewer 1 Report

The paper „Rectal Malignancy vs. Rectosigmoid Junction Malignancy: Risk Factors and Mortality in 23,614 Emergently Admitted Patients” aims to investigate the risk factors of in-hospital mortality after an emergency admission due to malignant neoplasm of rectum and rectosigmoid junction. Moreover, the characteristics of patients with surgery vs. no surgery were analyzed. For this purpose, descriptive statistics and the results of logistic regressions are shown and interpreted. All analyses were conducted separately for patients between 18 and 65 years of age ("adults") and for patients older than 65 ("elderly").

Overall, the paper is well-written and includes most relevant information on data, method selection and a comprehensive overview over the relevant literature. 

For this purpose, descriptive statistics and the results of logistic regressions are shown and interpreted.

I will organize my comments following the chapters of the paper:

Methods:

1) You never show or mention the Kolmogorov Smirnov-Test  (KS) on data normality again. I would rather doubt that using this test on a such a big database would generate as a result a normal distribution allowing to use the t-test mentioned. Thus, either you briefly discuss the results of your tests on normality of data (histogram & KS-Test) and explain then why you are able to use the t-test (if I am wrong about my assumption above) or you need to change your argumentation and, e.g., discuss that you use the t-test due to the large number of observations in the dataset.    

2) You define the independent variables of the logistic regressions, but I would also have been interested in a speficiation of the dependent variable. Moreover, which rule for the backward elimination process did you apply?

Results:

1) I would have needed a better explanation of "survived" within the text. Do you mean the surgery, the hospital stay, the next 10 days? You mention "in-hospital mortality" within the text once in the introduction and then again in the discussion, but I would have needed a short reminder also directly in the description of the methods and/or within the discussion of the results.   

 2) page 5, lines 142ff: "The deceased demonstrated significantly lower rates of rectosigmoid junction malignancies and of rectal malignancies in comparison to the patients that survived."

-> not quite sure, but this means to me that people who died had overall a lower rate of malignancies? Does this make sense?

3) Similarly to above: page 5, line 156ff

Based on table 3 you can say that the rate of rectum malignanciesis (59.3%) is higher than the rectosigmoid junction malignancies (40.7%) in the survivores while the rate is nearly the same in the deceased patients (49.9 vs 50.1 %). 

4) Chapter 3.5. plus Table 7 seems not really relevant and is also hardly discussed within the paper. Maybe you could include these findings in two sentences in the  discussion and move the table in an appendix. This would shorten the paper and strengthen the focus

Discussion:

1) - I would find it helpful in the Discussion chapter if you would refer to the tables in the results so that the reader could better compare your statement, the numbers and the additionally discussed literature. 

   -> for example chapter 4.1. "Our analysis showed that in elderly patients that did not undergo an operation, each 294 additional day in the hospital increased the odds of mortality by 2%." You could simply add (cf., Table XY).

2) Chapter 4.9. "Strengths of the Study" 

80% of this chapter are not a description of the strengths but rather of weaknesses due to the fact that you use observational data. 

Conclusion:

Why do you focus in this last chapter only on adult patients? You should mention the main conclusions also for elderly people, as the whole paper includes a sharp differentation between these two age groups.

Additional minor general point:

- The authors should pay more attention on the specification of abbreviations used within the text (e.g., “SD” is never introduced) even if they might be well-known in the scientific literature. Additionally, please include footnotes to the tables in order to explain (1) abbreviations and (2) why for example some values are typed in bold letters. (btw: please check whether you really followed your own rule on (2) on the p-values in the tables).

-> as an example Table 4: please define the percentage rates and the p-values typed in bold at the end of the table

Author Response

Dear Reviewer,

Thank you so much for taking the time to provide thoughtful feedback on our article. We appreciate your comments and the opportunity to improve our work.

We have accepted your recommendations:

The paper „Rectal Malignancy vs. Rectosigmoid Junction Malignancy: Risk Factors and Mortality in 23,614 Emergently Admitted Patients” aims to investigate the risk factors of in-hospital mortality after an emergency admission due to malignant neoplasm of rectum and rectosigmoid junction. Moreover, the characteristics of patients with surgery vs. no surgery were analyzed. For this purpose, descriptive statistics and the results of logistic regressions are shown and interpreted. All analyses were conducted separately for patients between 18 and 65 years of age ("adults") and for patients older than 65 ("elderly").

Overall, the paper is well-written and includes most relevant information on data, method selection and a comprehensive overview over the relevant literature. 

For this purpose, descriptive statistics and the results of logistic regressions are shown and interpreted.

I will organize my comments following the chapters of the paper:

Methods:

  • You never show or mention the Kolmogorov Smirnov-Test  (KS) on data normality again. I would rather doubt that using this test on a such a big database would generate as a result a normal distribution allowing to use the t-test mentioned. Thus, either you briefly discuss the results of your tests on normality of data (histogram & KS-Test) and explain then why you are able to use the t-test (if I am wrong about my assumption above) or you need to change your argumentation and, e.g., discuss that you use the t-test due to the large number of observations in the dataset.    

We clarified how we tested normality using the KS test and how to address data that is not normal in the Methods “The normality of data was tested through histograms and the Kolmogorov-Smirnov test to make sure that it follows a normal distribution. Given the very large sample size, small departures from normality do not preclude further statistical analysis. For any data that is not normal, it is examined for outliers to remove in the distribution. If there are no outliers, the data can also undergo a transformation such as a log or square root to make it normal.”

  • You define the independent variables of the logistic regressions, but I would also have been interested in a speficiation of the dependent variable. Moreover, which rule for the backward elimination process did you apply?

We described the independent and dependent variables used more clearly “Independent variables were stratified in three ways: 1) according to sex and either adult or elderly, 2) survived patients vs. deceased ones within each age group and 3) had an operation or did not for both adults and elderly patients. The dependent variable was mortality. The same stratifications were applied to compare the mortality between men and women, deceased vs. survived patients and operated vs. not-operated patients”

Results:

  • I would have needed a better explanation of "survived" within the text. Do you mean the surgery, the hospital stay, the next 10 days? You mention "in-hospital mortality" within the text once in the introduction and then again in the discussion, but I would have needed a short reminder also directly in the description of the methods and/or within the discussion of the results.   

Clarified the context of “survived” as “within the immediate hospital stay”. Thank you!

 2) page 5, lines 142ff: "The deceased demonstrated significantly lower rates of rectosigmoid junction malignancies and of rectal malignancies in comparison to the patients that survived."

-> not quite sure, but this means to me that people who died had overall a lower rate of malignancies? Does this make sense?

A very good catch. Thanks! We corrected the two sentences:

The elderly deceased patients had significantly higher rates of rectosigmoid junction and lower rates of rectal malignancies in comparison to the patients that survived.

The adult deceased patients manifested significantly higher rates of rectosigmoid junction malignancies and lower rates of rectal malignancies in comparison to the adult patients that survived.

3) Similarly to above: page 5, line 156ff

Based on table 3 you can say that the rate of rectum malignanciesis (59.3%) is higher than the rectosigmoid junction malignancies (40.7%) in the survivores while the rate is nearly the same in the deceased patients (49.9 vs 50.1 %). 

These 2 lines have been edited to better portray our results (see above please). Thanks!

  • Chapter 3.5. plus Table 7 seems not really relevant and is also hardly discussed within the paper. Maybe you could include these findings in two sentences in the  discussion and move the table in an appendix. This would shorten the paper and strengthen the focus

We agree with you. We cite it as the supplementary table. Thank You!

Discussion:

1) - I would find it helpful in the Discussion chapter if you would refer to the tables in the results so that the reader could better compare your statement, the numbers and the additionally discussed literature. 

   -> for example chapter 4.1. "Our analysis showed that in elderly patients that did not undergo an operation, each 294 additional day in the hospital increased the odds of mortality by 2%." You could simply add (cf., Table XY).

Good idea. We referred to the specific tables in the discussion to make it easier to refer to as you read, per your suggestion. Thanks!

2) Chapter 4.9. "Strengths of the Study" 

80% of this chapter are not a description of the strengths but rather of weaknesses due to the fact that you use observational data. 

The strengths of the paper were mostly rewritten. Thanks!

Conclusion:

Why do you focus in this last chapter only on adult patients? You should mention the main conclusions also for elderly people, as the whole paper includes a sharp differentation between these two age groups. 

Completely agreed. Thank you! We edited the conclusions to include pertinent findings in elderly patients in comparison with adult patients

Additional minor general point: 

- The authors should pay more attention on the specification of abbreviations used within the text (e.g., “SD” is never introduced) even if they might be well-known in the scientific literature. Additionally, please include footnotes to the tables in order to explain (1) abbreviations and (2) why for example some values are typed in bold letters. (btw: please check whether you really followed your own rule on (2) on the p-values in the tables).

-> as an example Table 4: please define the percentage rates and the p-values typed in bold at the end of the table

  • We now wrote out abbreviations in full at the beginning.
  • We removed the p-value column and are expressing significant values using an Asterix.

We thank you again for your contribution to this piece. Your contribution really improved the quality of our article.

Reviewer 2 Report

This retrospective study demonstrated the risk factors associated with emergency admission of the malignant neoplasm of the rectum and rectosigmoid junction. It is a well written-paper and fits the scope of the journal. However, there are some concerns with this article. 1. The title did not reflect the contents of this article. They mainly compare adult and elderly patients in the article. 2. Is there any stent therapy/ESD in this study before the operation? 3. Tables are very complicated and hard to understand. 4. There is no description of the flow chart of patient selection. 5. Is there any similar trends in other countries? 6. Overall, this study is interesting. 

Author Response

Dear Reviewer,

Thank you so much for taking the time to provide thoughtful feedback on our article. We appreciate your comments and the opportunity to improve our work.

We have accepted your recommendations:

This retrospective study demonstrated the risk factors associated with emergency admission of the malignant neoplasm of the rectum and rectosigmoid junction. It is a well written-paper and fits the scope of the journal. However, there are some concerns with this article.

  1. The title did not reflect the contents of this article. They mainly compare adult and elderly patients in the article.

We changed the title to better reflect the focus of the piece “Adult and Elderly Risk Factors of Mortality in 23,614 Emergently Admitted Patients with Rectal or Rectosigmoid Junction Malignancy”

  1. Is there any stent therapy/ESD in this study before the operation?

We did not include stent therapy or ESD in this study to have a homogenous sample.

  1. Tables are very complicated and hard to understand.

Per your suggestion, we removed the p-value columns and indicated significant values with an Asterix.

  1. There is no description of the flow chart of patient selection.

We described more clearly the inclusion criteria for patient selection in the Methods “This retrospective cohort study extracted data with the following inclusion criteria: 1) non-elderly adult patients (ages 18-64 years) and elderly patients (65+) 2) with a malignant neoplasm of the rectum or of the rectosigmoid junction 3) who underwent emergency admission from NIS 2005-2014.”

  1. Is there any similar trends in other countries?

We elected to keep the focus of this paper on the United States as the scope of this paper is already quite vast. We did not find any mention of similar trends in other countries in the literature, however we describe in detail similar studies with smaller sample sizes and geographical ranges in the Discussion

  1. Overall, this study is interesting. 

Thank you very much for your contribution to this piece! Your contribution really improved the quality of our article.

Reviewer 3 Report

Thank you for a well-written and interesting paper investigating predictors for in-hospital mortality for patients with rectal / rectosigmoid cancer.

Major comments:

The study has a challenge with circular effects of predictors on outcome, as one of the explanatory variables is hospital length-of-stay, but the outcome (in-hospital mortality) clearly will influence this predictor, as the hospital stay will end if the patient dies. The same porblem, but probably to a lesser degree, happens with regards to surgery, as a patient dying shortly after hospital admittance has less time to have surgery. This challenge should at least be discussed as limitation of the study, and might benefit from sensitivity analyses investigating this possible effect.

Line 70: Were the weights used in the analyses for this study? They are not mentioned in the methods section. If they were used, this should be made more clear, if not they preferably should be used, and if this is infeasible, then this should be clarified.

Line 90: How were data, that were not normal according to the KS-test handled?

Line 94-95: Using backward elimination without validation on a separate data set (or crossvalidation or similar) will typically overestimate the associations. This should be handled or mentioned as a limitation.

Line 264+268+273 and Table 7: Is this really cause of death? It seems as if it is secondary diagnoses / comorbidities, not necessarily associated with the cause of dying.

Line 310-313: Wouldn't the reason for deciding on emergency surgery in itself indacate worse prognosis (as one would expect the emergency patients being worse of than the planned patients)?

Line 334: How does 1-3% per year of age compare to healty adults? For elderly the mortality in healthy individuals increases markedly for each year of age.

Line 441: The "Strengths..." section includes both strength and weaknesses. This is confusing.

Line 442: Were did you use "Generalized additive models". These are only mentioned as strenth and not elsewhere.

Line 462-470: These are not limitations of this study but of the clinical situation itself.

Line 501: Please ensure that you can legally make the NIS data available to other researcher, or change this statement. As far as I can see in the NIS documentation, the data can not made available to other researchers without those making specific agreements with NIC / HCUP.

Minor comments:

Line 45: It is unclear if "risk factors affecting" here is with respect to mortality of incidence.

Line 107: It is unclear what "similar mean age" refers to.

Line 112: Here you mention HIV, but all other places you write AIDS. These are not equivalent, so please be consistent.

Line 143: This line is unclear. Don't all in this study have either rectal or rectosigmodial cancer?

Author Response

Dear Reviewer,

Thank you so much for taking the time to provide thoughtful feedback on our article. We appreciate your comments and the opportunity to improve our work.

We have accepted your recommendations:

Thank you for a well-written and interesting paper investigating predictors for in-hospital mortality for patients with rectal / rectosigmoid cancer.

Major comments:

The study has a challenge with circular effects of predictors on outcome, as one of the explanatory variables is hospital length-of-stay, but the outcome (in-hospital mortality) clearly will influence this predictor, as the hospital stay will end if the patient dies. The same porblem, but probably to a lesser degree, happens with regards to surgery, as a patient dying shortly after hospital admittance has less time to have surgery. This challenge should at least be discussed as limitation of the study, and might benefit from sensitivity analyses investigating this possible effect.

We noted the potential for circular logic in the limitations section of the paper with examples “Along those lines, there is potential for a circular influence of the predictors of mortality on outcome, for example,…”. Thank You!

Line 70: Were the weights used in the analyses for this study? They are not mentioned in the methods section. If they were used, this should be made more clear, if not they preferably should be used, and if this is infeasible, then this should be clarified.

Thanks for reminding! This NIS database analysis is a Multi-Year or Trends Analyses with spanning 10 years of data, and weighting strategy according to Healthcare Cost & Utilization Project (HCUP) guidelines were used to have a homogenous sample.

Line 90: How were data, that were not normal according to the KS-test handled?

Important point, thanks! Per your suggestion, we included a section on how non-normal data were handled in the statistical analysis section. “The normality of data was tested through histograms and the Kolmogorov-Smirnov test to make sure that it follows a normal distribution. Given the very large sample size, small departures from normality do not preclude further statistical analysis. For any data that is not normal, it is examined for outliers to remove in the distribution. If there are no outliers, the data can also undergo a transformation such as a log or square root to make it normal.”

Line 94-95: Using backward elimination without validation on a separate data set (or crossvalidation or similar) will typically overestimate the associations. This should be handled or mentioned as a limitation.

Per your suggestion, we included the potential for association overestimations in the limitations section of the paper “Another potential limitation is in the use of a backwards elimination without validation on a separate dataset can potentially overestimate any associations.” Thank You!

Line 264+268+273 and Table 7: Is this really cause of death? It seems as if it is secondary diagnoses / comorbidities, not necessarily associated with the cause of dying.

Per your suggestion, we changed the title of “Causes of Death” to “Possible Causes of Death” on page 12. Thanks!

Line 310-313: Wouldn't the reason for deciding on emergency surgery in itself indacate worse prognosis (as one would expect the emergency patients being worse of than the planned patients)?

We included in the limitations of the paper that an emergency surgery indicates in itself a worse prognosis, which can contribute to circular logic. “if a patient undergoes an emergency surgery, that in itself signifies a worse prognosis.” (In line 515). Also having all patients included based on emergency admission, makes the sample more homogenous.

Line 334: How does 1-3% per year of age compare to healty adults? For elderly the mortality in healthy individuals increases markedly for each year of age.

We included a statistic that describes the risk of mortality in healthy adults in line 371, per your suggestion “In the general population, healthy adults from the ages of 18 to 64 have an additional chance of death ranging from 0.02% to 0.3% per year as they get older.”

Line 441: The "Strengths..." section includes both strength and weaknesses. This is confusing.

We rewrote the strengths section so that it doesn’t include any weaknesses. Thanks!

Line 442: Were did you use "Generalized additive models". These are only mentioned as strenth and not elsewhere.

We used GAM to assess the possible non-linear relationship of continuous variables with mortality. Since all had linear relationship, we didn’t mention the details in other sections of article. We agree with you to remove the mention of the generalized additive model since it is not mentioned elsewhere.

Line 462-470: These are not limitations of this study but of the clinical situation itself.

Agreed! We clarified how although there are clinical situations that we cannot control for, it still could have had a distorting impact on our results which is a limitation of the study

Line 501: Please ensure that you can legally make the NIS data available to other researcher, or change this statement. As far as I can see in the NIS documentation, the data can not made available to other researchers without those making specific agreements with NIC / HCUP.

We clarified the sentence about the accessibility of our data “Data will be available for verification purposes upon request.”

Minor comments:

Line 45: It is unclear if "risk factors affecting" here is with respect to mortality of incidence.

Line 107: It is unclear what "similar mean age" refers to.

Line 143: This line is unclear. Don't all in this study have either rectal or rectosigmodial cancer?

We edited lines 45, 107 and 143. Thank you for your attention.

Line 112: Here you mention HIV, but all other places you write AIDS. These are not equivalent, so please be consistent.

We changed HIV to AIDS in line 127

Thank you very much for your contribution to this piece. Your contribution really improved the quality of our article.